# *FOXD1* Repression Potentiates Radiation Effectiveness by Downregulating *G3BP2* Expression and Promoting the Activation of *TXNIP*-Related Pathways in Oral Cancer

**DOI:** 10.3390/cancers12092690

**Published:** 2020-09-21

**Authors:** Che-Hsuan Lin, Hsun-Hua Lee, Wei-Min Chang, Fei-Peng Lee, Lung-Che Chen, Long-Sheng Lu, Yuan-Feng Lin

**Affiliations:** 1Department of Otolaryngology, School of Medicine, College of Medicine, Taipei Medical University, Taipei 11031, Taiwan; cloudfrank@gmail.com (C.-H.L.); fplee@tmu.edu.tw (F.-P.L.); b101093017@tmu.edu.tw (L.-C.C.); 2Department of Otolaryngology, Taipei Medical University Hospital, Taipei Medical University, Taipei 11031, Taiwan; 3Graduate Institute of Clinical Medicine, College of Medicine, Taipei Medical University, Taipei 11031, Taiwan; kaorulei@yahoo.com.tw; 4Department of Neurology, Shuang Ho Hospital, Taipei Medical University, New Taipei City 235, Taiwan; 5Department of Neurology, School of Medicine, College of Medicine, Taipei Medical University, Taipei 11031, Taiwan; 6Department of Neurology, Vertigo and Balance Impairment Center, Shuang Ho Hospital, Taipei Medical University, New Taipei City 235, Taiwan; 7School of Oral Hygiene, College of Oral Medicine, Taipei Medical University, Taipei 11031, Taiwan; weiminchang@tmu.edu.tw; 8Department of Otolaryngology, Shuang-Ho Hospital, Taipei Medical University, New Taipei City 235, Taiwan; 9Department of Radiation Oncology, Taipei Medical University Hospital, Taipei Medical University, Taipei 11031, Taiwan; lslu@tmu.edu.tw; 10Graduate Institute of Biomedical Materials and Tissue Engineering, College of Biomedical Engineering, Taipei Medical University, Taipei 11031, Taiwan; 11Cell Physiology and Molecular Image Research Center, Wan Fang Hospital, Taipei Medical University, Taipei 11696, Taiwan

**Keywords:** radiotherapy, *FOXD1*, *G3BP2*, *TXNIP*, oral cancer

## Abstract

**Simple Summary:**

Radioresistance remains a critical issue in treating oral cancer patients. This study was thus aimed to identify a potential drug target for enhancing the therapeutic effectiveness of irradiation and uncover a possible mechanism for radioresistance in oral cancer. Here we show that *FOXD1*, a gene encoding forkhead box d1 (Foxd1), is significantly upregulated in primary tumors compared to normal tissues and serves as a poor prognostic marker in oral cancer patients receiving radiotherapy. *FOXD1* repression by a gene knockdown experiment dramatically enhanced the cytotoxic efficacy of irradiation probably via activating the p53-related DNA repairing pathways and reinforcing the T cell-mediated immune responses in oral cancer cells. Our findings demonstrate that *FOXD1* may play a pivotal role in conferring radioresistance, which might provide a new strategy to combat the irradiation-insensitive oral cancer cells via therapeutically targeting *FOXD1* activity.

**Abstract:**

Radiotherapy is commonly used to treat oral cancer patients in the current clinics; however, a subpopulation of patients shows poor radiosensitivity. Therefore, the aim of this study is to identify a biomarker or druggable target to enhance the effectiveness of radiotherapy on oral cancer patients. By performing an in silico analysis against public databases, we found that the upregulation of *FOXD1*, a gene encoding forkhead box d1 (Foxd1), is extensively detected in primary tumors compared to normal tissues and associated with a poor outcome in oral cancer patients receiving irradiation treatment. Moreover, our data showed that the level of *FOXD1* transcript is causally relevant to the effective dosage of irradiation in a panel of oral cancer cell lines. The *FOXD1* knockdown (FOXD1-KD) dramatically suppressed the colony-forming ability of oral cancer cells after irradiation treatment. Differentially expressed genes analysis showed that *G3BP2*, a negative regulator of p53, is predominantly repressed after FOXD1-KD and transcriptionally regulated by Foxd1, as judged by a luciferase-based promoter assay in oral cancer cells. Gene set enrichment analysis significantly predicted the inhibition of E2F-related signaling pathway but the activation of the interferons (IFNs) and p53-associated cellular functions, which were further validated by luciferase reporter assays in the FOXD1-KD oral cancer cells. Robustly, our data showed that FOXD1-KD fosters the expression of *TXNIP*, a downstream effector of IFN signaling and activator of p53, in oral cancer cells. These findings suggest that *FOXD1* targeting might potentiate the anti-cancer effectiveness of radiotherapy and promote immune surveillance on oral cancer.

## 1. Introduction

Oral squamous cell carcinoma (OSCC) is the most common (>90%) of all oral cancers [1]. In 2017, according to the Oral Cancer Foundation estimation, 49,750 Americans were diagnosed with oral and oropharyngeal cancer and 9750 individuals died from this cancer. Among those cancer patients, the 5 year survival rate was about 57% and disease-free survival rate was 58%. Risk factors, such as betel nut chewing, alcohol consumption, and tobacco smoking have been considered for oral cancer [2]. According to the National Comprehensive Cancer Network (NCCN) guidelines, the treatment options are varied based on different stages, including surgery, radiotherapy, chemotherapy, and target therapy (e.g., cetuximab, bevacizumab) [3]. External beam radiotherapy (EBRT) is generally employed in three situations, such as adjuvant to primary surgery to enhance loco-regional control for advanced cases or cases with unfavorable pathological features, primary treatment for medically inoperable cases and salvage or palliative treatment for the persistent or recurrent disease [4]. Modern radiotherapy is most frequently used to treat advanced head and neck cancer. However, there were no clinical studies to address the useful biomarker for predicting the effectiveness of radiotherapy, which could offer an indication for a therapeutic recommendation [5].

The forkhead box (FOX) family consists of various tissue and cell type-specific transcription regulators with a conserved winged-helix DNA-binding domain (DBD) or forkhead domain [6]. FOX family members are comprised of a common DBD adjacent to distinct transactivation and repression domains [7]. FOX family members are thought to be important regulators in physiological development during embryogenesis [7,8,9]. In addition, FOX transcription factors have been shown to associate with cancer research such as drug resistance, tumor growth, genomic alterations or drivers of initiation and thought to be a potential therapeutic strategy to combat cancer and putative biomarkers for specific cancers [6,10]. The FOXD subfamily includes *FOXD1*, *FOXD2*, *FOXD3* and *FOXD4* and functions as a critical regulator in normal cell development and disease progression [11,12,13,14,15,16,17,18]. Moreover, the FOXD subfamily has also been found to play a role as an important regulator for tumorigenesis and cancer progression [19,20,21,22,23,24], e.g., therapeutic resistance and cancer metastasis, and serves as a prognostic biomarker in several types of cancer [25,26,27]. In oral squamous cell carcinoma, the upregulation of FOXM1 has been shown to correlate with tumor growth [28], and FOXP3-mediated immune modulation appeared to potentiate the anti-tumor effectiveness of regulatory T cell-based immunotherapy [29]. In addition, *FOXD1* upregulation has been shown to promote therapeutic resistance in breast cancer [24]; however, its role in conferring therapeutic resistance, e.g., radioresistance, in OSCC remains unknown.

Although previous reports have shown that FOXD subtypes (FOXDs) play a critical role in the mechanism for tumorigenesis and cancer progression, their roles in regulating oral cancer development and conferring the radiation resistance of oral cancer remain largely unknown. The aim of this study was thus focused on dissecting the transcriptional profiling of FOXDs in normal tissues and primary tumors and evaluating their clinical relevance in oral cancer. Our data demonstrate that the upregulation of *FOXD1* compared to other FOXDs is extensively detected in primary tumors and significantly correlated with a poorer clinical outcome in oral cancer patients. Our results further show that that *FOXD1* upregulation correlates with a poor responsiveness of oral cancer patients to radiotherapy and desensitizes oral cancer cells to irradiation treatment probably via elevating the *G3BP2* and E2F-related pathways and suppressing the signaling cascades related to the *TXNIP*-associated interferon responsiveness and p53 activity. This study is the first to document the oncogenic role of *FOXD1* in oral cancer.

## 2. Results

### 2.1. FOXD1 Upregulation Is Dominant for Primary Tumors Compared to Normal Tissues Derived from Patients with Oral Cancer

We firstly dissected the transcriptional profile of genes encoding forkhead box d (Foxd) protein family in normal tissues and primary tumors derived from The Cancer Genome Atlas (TCGA) head and neck cancer patients (Figure 1A). The data showed that the mRNA levels of *FOXD1*, *FOXD2*, *FOXD3* and *FOXD4* in primary tumors are significantly (*p* < 0.001) higher than that of normal tissues (Figure 1B). Moreover, except *FOXD3*, we found that *FOXD1*, *FOXD2* and *FOXD4* are more significantly (*p* < 0.01) upregulated in primary tumor tissue compared to normal adjacent tissues derived from TCGA head and neck cancer patients (Figure 1C). We further analyzed the transcriptional profile of *FOXD1*, *FOXD2* and *FOXD4* in the anatomic subdivision of TCGA head and neck cancer (Appendix A) and found that the expression of *FOXD2* in the hypopharynx and *FOXD4* in the tonsil is relatively higher than other tissues, such as the oral cavity (Appendix A). However, the mRNA levels of *FOXD1* among the anatomic subdivision of head and neck cancer tissues appeared to show no difference (Appendix A). Robustly, *FOXD1*, but not *FOXD2*, *FOXD3* and *FOXD4*, was significantly (*p* < 0.01) upregulated in primary tumors compared to normal adjacent tissues derived from oral cancer patients deposited in the GSE42743 dataset (Figure 1D). Similar views were also found in the paired normal adjacent tissue and primary tumor derived from TCGA oral cancer subjects (Appendix A).

### 2.2. FOXD1 Upregulation Predicts a Poor Prognosis in Oral Cancer Patients

We next evaluated the prognostic significance of *FOXD1*, *FOXD2*, *FOXD3* and *FOXD4* in TCGA head and neck cancer patients. Kaplan–Meier analysis revealed that *FOXD1*, as compared to *FOXD2*, *FOXD3* and *FOXD4*, upregulation more significantly (*p* = 0.008) predicts a poor overall survival rate in TCGA head and neck cancer patients (Figure 2A). Moreover, under the condition of overall survival probability, the Cox regression test using univariate and multivariate modes revealed that *FOXD1* serves as an independent risk factor in comparison with other clinical parameters for predicting the prognosis of head and neck cancer patients (Figure 2B). Although the mRNA levels of *FOXD1* showed no differences between the groups of age (<0 vs. ≥60), gender (female vs. male) and pathologic staging (T1/2 vs. T3/4; N0 vs. N1; stage I/II vs. II/IV), *FOXD1* expression was relatively higher in primary tumors derived from head and neck cancer patients without smoking history or oral squamous cell carcinoma (OSCC) subdivision (Figure 2C). Importantly, a higher mRNA level of *FOXD1* was detected in patients who died from OSCC compared to patients who died from other causes and who were alive at the end of follow-up (Figure 2D).

### 2.3. FOXD1 Repression Enhances the Therapeutic Responsiveness of Oral Cancer to Radiotherapy

Since radiotherapy is commonly used to treat oral cancer patients, we next dissected the correlation between *FOXD1* expression and irradiation responsiveness in oral cancer. By using TCGA head and neck cancer database, we found that *FOXD1* expression significantly (*p* = 0.036) correlates with a shorter time to new tumor event in patients receiving radiotherapy (Figure 3A). A similar view was also found in GSE42743 OSCC patients receiving post-operative radiotherapy (Figure 3B). Moreover, our data showed that the endogenous protein and mRNA levels of *FOXD1* are causally associated with the cell viability and colony-forming ability detected after 24 h post-treatment with irradiation at 8 Gy in oral cancer cell lines HSC2, HSC3, HSC4 and SAS (Figure 3C–E and Appendix A). To understand if *FOXD1* repression could enhance the radiosensitivity, we next performed *FOXD1* knockdown experiments in HSC4 cells. Robustly, *FOXD1* knockdown by its two independent shRNA clones (Figure 3F) significantly (*p* < 0.01) suppressed the colony-forming ability of HSC4 cells pretreated with irradiation at 4 or 8 Gy (Figure 3G,H) as well as cell viability (Appendix A).

### 2.4. FOXD1 Repression Results in the Downregulation of G3BP2-Related Pathway in Oral Cancer

To delineate a possible mechanism for the *FOXD1*-associated radioresistance in oral cancer, we next analyzed the mRNA levels of annotated genes on Illumina microarray after *FOXD1* knockdown in A375 and MeWo melanoma cells by using GSE111766 dataset (Figure 4A). The obtained fold changes of these annotated genes in the *FOXD1*-silencing cells compared to control cells were further used to perform differentially expressed genes (DEG) and gene set enrichment analysis (GSEA) experiments (Figure 4B,C and Appendix A). DEG experiments revealed that *G3BP2* expression is predominantly downregulated after *FOXD1* knockdown in both detected cell lines (Figure 4B). We next validated this finding in the HSC4 cells and found that *FOXD1* knockdown dramatically reduces the protein and mRNA levels in HSC4 cells (Figure 4D). Similar to *FOXD1*, the mRNA levels of *G3BP2* in HSC4 cells were higher than other oral cancer cell lines (Appendix A). Since the Foxd1 protein acts as a transcription factor, we thus performed luciferase-based promoter activity assay to examine if *G3BP2* expression is transcriptionally regulated by Foxd1. By using in silico analysis, we found that *G3BP2* promoter region (−740 to −734) contains Foxd1 DNA-binding sequences GTAAACA (Figure 4E). Luciferase-based promoter activity assays were performed in HSC4 cells by transfecting an expression vector containing Gaussia luciferase gene adjacent to the *G3BP2* promoter (−1301 to +141) harboring wild-type or mutated (GTAAACA to GTCCCCA, Figure 4E) Foxd1 DNA-binding sequences. The data demonstrated that luciferase activity by transfecting the vector-containing wild-type, not mutant, Foxd1 DNA-binding sequences is dramatically reduced in the *FOXD1*-silencing cells compared to parental and non-silencing control cells (Figure 4F). Moreover, we found that the gene expression of *FOXD1* and *G3BP2* is positively correlated in primary tumors derived from GSE42734 OSCC patients (Figure 4G). These findings indicate that *G3BP2* is one of downstream target genes for Foxd1 transcription factor.

### 2.5. FOXD1 Repression Promotes Interferon Responsiveness and p53-Related DNA Repairing Pathway in Oral Cancer

GSEA results show that the gene sets perturbed upon interferon-alpha (IFN-α) or IFN-γ stimulation are positively correlated with the altered gene expression after *FOXD1* knockdown in A375 cells (Figure 5A). To validate if the IFN-α and IFN-γ-related signaling pathways are activated after *FOXD1* knockdown in oral cancer cells, we performed luciferase-based promoter activity assays using commercialized luciferase-containing vectors used for determining the activities of IFN-α and IFN-γ-related signaling pathways in HSC4 cells (Figure 5B,C). The data showed that luciferase activities are significantly (*p* < 0.01) increased after *FOXD1* knockdown (Figure 5B,C), indicating a negative correlation between *FOXD1* expression and interferon actions in oral cancer. Since interferons play a critical role in triggering T cell-mediated tumor-killing effect, we were interested in dissecting the expression of PD-L1 which is able to suppress T cell function via interacting with its receptor PD-1 in oral cancer. By using GSE42734 dataset, we found that the mRNA levels of *FOXD1* and PD-L1-coding gene *CD274* are positively correlated in primary tumors derived from oral cancer patients (Figure 5D). Robustly, *FOXD1* knockdown resulted in the repression of PD-L1 gene in HSC4 cells (Figure 5E).

In addition, GSEA simulation also predicted the inactivation of the E2F-related signaling axis but the activation of p53-associated pathway in response to *FOXD1* knockdown (Figure 5F). *FOXD1* knockdown in HSC4 cells led to the reduction of phosphorylated Rb (Figure 5G), which is incapable of inhibiting E2F activity. Luciferase-based promoter activity assay confirmed that E2F is suppressed in HSC4 cells after *FOXD1* knockdown (Figure 5H). Intriguingly, we found that *TXNIP*, a gene that encodes thioredoxin-interacting protein, is a consensus gene in the upregulated genes after *FOXD1* knockdown in the melanoma cells (Figure 4B) and the gene sets of IFN-α/γ responses and p53 pathway (Figure 5I). Moreover, *FOXD1* knockdown appeared to elevate the expression of *TXNIP* in HSC4 cells (Figure 5J). In oral cancer tissues derived from the GSE42734 dataset, we found that the expression of *FOXD1* and *TXNIP* is significantly (*p* = 0.029) inversed (Figure 5K).

## 3. Discussion

Recent reports indicated that *FOXD1* plays a oncogenic effect in several types of cancer [30,31,32,33,34,35,36] and likely associates with the mechanism for radioresistance [37]. In this study, our data showed that *FOXD1* is capable of directly regulating the expression of *G3BP2*, which is capable of inhibiting p53 activity through a direct binding, which may further promote p53 nuclear export via increasing p53 sumoylation [38] and serves as a poor prognostic marker in prostate cancer patients [39]. Tumor suppressor p53 acts as a critical regulator for DNA damage response and controls the G1 checkpoint of cell cycle via interacting with Rb-E2F pathway [40]. The inactive mutation of p53 was also detected in lung cancer with radioresistance [41]. In addition, our results reveal that *FOXD1* upregulation is accompanied with an enhanced activity of E2F, probably due to the dissociation with hyper-phosphorylated Rb protein, which has been shown to promote cell cycle progression and desensitize oral cancer cells to irradiation [42]. Here, we firstly document that *FOXD1* upregulation probably associates with the mechanism for radioresistance in oral cancer, probably via activating *G3BP2* and E2F-related pathways and negatively regulating the p53-related cellular functions (Figure 6). Therefore, the therapeutic targeting of *FOXD1* might be a new strategy to potentiate the efficacy of irradiation in treating oral cancer.

Radiation therapy has been shown to enhance the anti-tumor capacity of adaptive immunity by augmenting a type I interferon (IFN)-dependent innate immune sensing of tumors [43]. Based on this immunomodulatory effect of radiation therapy, several clinical trials were performed to evaluate the anti-cancer effectiveness of combining irradiation with IFNs [44,45,46]. Similarly, our results demonstrate that *FOXD1* knockdown enhances the radiosensitivity of oral cancer cells via activating the IFN-α and IFN-γ-responsive pathways. In addition, the immunostimulatory effects of radiation therapy, such as improved immune cell recruitment and enhanced susceptibility to T cell-mediated cell death, were also reported previously [47]. Here, we found that *FOXD1* knockdown is concurrently accompanied with a reduced expression of PD-L1, a critical suppressor for T cell function through the binding with PD-1, in oral cancer cells with poorer radiosensitivity. These findings suggest that *FOXD1* repression may not only promote the therapeutic efficacy of irradiation but also potentiate the T cell-mediated adaptive immunity in combating oral cancer.

In contrast, *TXNIP*, a member of the tumor suppressor family, has been shown to increase p53 stability and activity, thereby sensitizing breast cancer cells to apoptotic stimulation [48]. On the other hand, a recent report showed that an increased level of *TXNIP* probably results from the activation of the JAK-STAT pathway and is associated with a favorable prognosis in patients with renal cell carcinoma [49]. In this study, our data showed that *FOXD1* knockdown dramatically enhances the expression of *TXNIP* and the IFN-α/γ responsiveness, which is determined by the interaction of the JAK/STAT1 signaling axis with IFN-stimulated response element (ISRE) and IFN-gamma activation site (GAS) response element within the upstream promoter of the luciferase gene in oral cancer cells. Based on these findings, we thought that *FOXD1* upregulation confers radioresistance by downregulating the JAK-STAT pathway-mediated *TXNIP* expression, which may ultimately decrease and inactivate p53 in oral cancer (Figure 6).

## 4. Materials and Methods

### 4.1. Data Collection and Processing from TCGA and GEO Databases

The clinical data and overall survival (OS) time for TCGA head and neck cancer patients were collected from the UCSC Xena website (UCSC Xena. Available online: http://xena.ucsc.edu/welcome-to-ucsc-xena/). The molecular data obtained by RNAseq (polyA þ Illumina HiSeq, CA, USA) analysis of the TCGA head and neck cancer cohort were also downloaded from the UCSC Xena website. Microarray results with accession numbers GSE42743, which was performed by Holsinger C et. al. from Stanford University School of Medicine to compare differences of gene expression between oral cancer samples and adjacent normal mucosa, and GSE111766, which was established by Larribère L et. al. from DKFZ Research Center in Germany to dissect the alteration of gene expression after *FOXD1* knockdown in melanoma cell lines, and the related clinical data were obtained from the Gene Expression Omnibus (GEO) database on the NCBI website. The raw intensities of mRNA levels derived from GSE42743 dataset were normalized by robust multichip analysis using GeneSpring GX11 (Agilent Technologies, CA, USA). The mRNA expression levels were normalized by the median of the detected samples and presented as log_2_ values. The fold changes of gene expression after *FOXD1* knockdown in A375 [*FOXD1* knockdown #1 (GSM3039523, GSM3039524, GSM3039525) versus control (GSM3039516, GSM3039517, GSM3039518) and MeWo (*FOXD1* knockdown #1 (GSM3039519, GSM3039520) versus control (GSM3039514, GSM3039515) were obtained by using GEO2R software and presented as log_2_ values.

### 4.2. Cell Culture Condition

Oral cancer cell lines HSC-2, HSC-3, HSC-4 and SAS were obtained from the Japanese Collection of Research Bioresources (JCRB) Cell Bank and were cultivated in Dulbecco’s modified Eagle’s medium (DMEM) supplemented with 10% fetal bovine serum (FBS) and 1% non-essential amino acids (NEAA) at 37 °C in a humidified atmosphere containing 7% CO2. 293T cells were obtained from the American Type Culture Collection (ATCC) and cultivated in DMEM containing 10% FBS and incubated at 37 °C with 5% CO2. The cell lines used in this study were routinely subjected to short tandem repeat (STR) analysis, morphologic and growth characteristics and mycoplasma detection.

### 4.3. Radiation Exposure and Cell Viability Analysis

Cells were exposed to 6 MV X-rays using a linear accelerator (Digital M Mevatron Accelerator, Siemens Medical Systems, CA, USA) at a dose rate of 8 Gy/min. To ensure electronic equilibrium, a tissue-equivalent bolus (2 cm) was placed on the top of the plastic tissue-culture flasks. To obtain full backscatter, tissue-equivalent material (10 cm) was placed under the flasks. After the exposure to the designated irradiation doses, cells were centrifuged and resuspended in PBS. For the cell viability assay, equal volumes of cell suspension and Trypan blue solution (0.4% in PBS) were mixed in order to stain the dead cells that were then placed on a hemocytometer and counted under a microscope.

### 4.4. Colony Formation Assay

After radiation exposure, cells (2000/well) were seeded on polystyrene 6-well plates and cultivated for 2 weeks. Cells were then fixed with 80% ethanol and stained with 1% crystal violet. After several washes, 30% acetic acid was used to solubilize the remaining crystal violet. The optical density of solubilized crystal violet was then measured by a photometer using 595 nm wavelength.

### 4.5. Lentivirus-Driven shRNA Infection

All shRNA plasmids with a puromycin selection marker were purchased from the National RNAi Core Facility Platform in Taiwan. Lentiviral particles were produced by cotransfecting the shRNA plasmid with the pMDG and p∆8.91 vectors into 293T cells using a calcium phosphate transfection kit (Invitrogen). Post-transfection for 48–72 h, the media were harvested as viral stocks. Cells grown with 50% confluence on 6-well plates were replenished with fresh media containing 5 µg/mL polybrene (SantaCruz, Dallas, TX, USA) prior to the infection with a lentiviral particle-driven control or target gene shRNA at 2–10 multiplicity of infection (MOI) overnight. To select cells that stably express the control or target gene shRNA, cells were further incubated with puromycin (10 µg/mL) for 24 h. To confirm the knockdown efficiency, cell lysates from the puromycin-resistant cells were subsequently subjected to RT-PCR analysis.

### 4.6. Reverse Transcription PCR (RT-PCR) and Quantitative PCR (Q-PCR)

A TRIzol extraction kit (Invitrogen) was employed to extract total RNA from the detected cells. Aliquots (5 µg) of total RNA were converted to cDNA by M-MLV reverse transcriptase (Invitrogen) and then amplified by PCR with Taq-polymerase (Protech) using paired primers (for *FOXD1*, forward-CTATGACCCTGAGCACTGAGATGTC and reverse-GCAGGATGTCATCGTCGTCCTC; for *G3BP2*, forward-CTGAAGAGCTGAAACCACAAGTGG and reverse-CGGTTGTCAGAGTCATTCTGTTCC; for PD-L1, forward-GCTGCACTTCAGATCACAGATGTG and reverse-GTGTTGATTCTCAGTGTGCTGGTC; for *TXNIP*, forward-GAGGTGTGTGAAGTTACTCGTGTC and reverse-GACATCCACCAGATCCACTACTTC; for *GAPDH*, forward-AGGTCGGAGTCAACGGATTTG and reverse-GTGATGGCATGGACTGTGGTC). Q-PCR was performed by using Power SYBR^TM^ Green PCR Master Mix (Thermo Fisher). The obtained mRNA level of the detected gene was further normalized to the mRNA level of GAPDH. The 2^−ΔΔCt^ method was used to calculate the fold changes.

### 4.7. Western Blotting Assay

Aliquots of total protein (20–100 µg) were loaded into each well of an SDS gel, separated by electrophoresis and then transferred to PVDF membranes. Prior to the incubation with primary antibodies against Foxd1 (Santa Cruz), G3BP2 (Abcam), total/phosphorylated Rb (Cell Signaling) and GAPDH (AbFrontier) overnight at 4 °C, the membranes were incubated with blocking buffer (5% skim milk in TBS containing 0.1% Tween-20) for 2 h at room temperature. After several washes, the membranes were further incubated with a peroxidase-labeled secondary antibody for another 1 h at room temperature. Immunoreactive bands were finally visualized by using an enhanced chemiluminescence system (Amersham Biosciences, Tokyo, Japan). Uncut blots can be found at Appendix A.

### 4.8. Site-Directed Mutagenesis and Luciferase Reporter Assay

PCR for site-directed mutagenesis against Foxd1 DNA-binding site within *G3BP2* promoter was performed with paired primers, forward-TTGCACACTAAACTCACCTACATTGGGGACTTAACCCATCCTAGGTTTTAAGC and reverse-GCTTAAAACCTAGGATGGGTTAAGTCCCCAATGTAGGTGAGTTTAGTGTGCAA, using a pfu polymerase kit (Stratagene, La Jolla, CA, USA). The PCR products were treated with Dpn1 endonuclease (New England BioLabs, Hitchin, Hertfordshire, UK) to digest the methylated parental DNA template.

Luciferase reporter vectors containing IFN-stimulated response element (ISRE) and IFN-gamma activation site (GAS) response element were purchased from Promega. Luciferase reporter vector containing E2F response elements within the promoter region was purchased from Addgene. Gaussia luciferase reporter vector containing *G3BP2* promoter was purchased from Gene Copoeia. Cells grown with 70% confluence on in 6-well plates were cotransfected with luciferase reporter and Renilla luciferase expression vectors. Post-transfection for 24 h, a Dual-Glo^®®^ Luciferase Assay System (Promega) was employed to measure the cellular luciferase activities. The assay procedure was performed by a protocol according to the manufactural guideline. The level of Gaussia and firefly luminescence was normalized to that of Renilla luminescence.

### 4.9. Statistical Analysis

SPSS 17.0 software (Informer Technologies, Roseau, Dominica) was used to analyse statistical significance. The paired t-test was utilized to compare Gαh gene expression in the cancer tissues and corresponding normal tissues. Pearson’s and non-parametric Spearman’s correlation tests were performed to estimate the association among mRNA levels of *FOXD1*, *G3BP2*, PL-D1 and *TXNIP* in the detected primary tumors. Evaluation of survival probabilities was determined by Kaplan–Meier analysis and log-rank test. Student t-test, paired t-test and one-way ANOVA with Tukey’s test were used to estimate the statistical significance of the detected gene expression in clinical samples. The non-parametric Friedman test was used to analyze the non-parametric data. *p* values < 0.05 in all analyses were considered statistically significant.

## 5. Conclusions

Our results demonstrate that *FOXD1* is highly expressed by primary tumors compared to the adjacent normal tissues and serves as a poor prognostic marker in oral cancer patients receiving irradiation therapy. The targeting of *FOXD1* might be a good strategy to enhance the radiosensitivity of oral cancer cells via downregulating *G3BP2*-related pathways and upregulating the *TXNIP*-associated cellular functions.

## Figures and Tables

**Figure 1 cancers-12-02690-f001:**
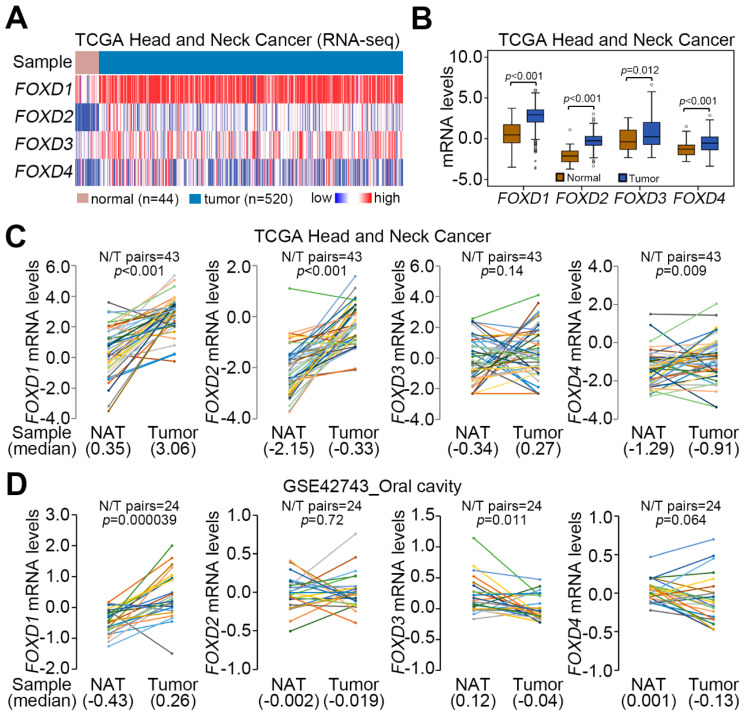
*FOXD1* is upregulated in primary tumors compared to normal tissues derived from head and neck cancer and oral cancer patients. (**A**) The heatmap for the transcriptional profiling of genes encoding *FOXD1*, *FOXD2*, *FOXD3* and *FOXD4* using the TCGA head and neck cancer database. (**B**) Boxplot for the mRNA levels of *FOXD1*, *FOXD2*, *FOXD3* and *FOXD4* in normal tissues and primary tumors derived from TCGA head and neck cancer database. The statistical differences were analyzed by student t-test. (C and D) The mRNA levels of *FOXD1*, *FOXD2*, *FOXD3* and *FOXD4* in the normal adjacent tissues (NAT) and primary tumors from the TCGA head and neck cancer patients (**C**) and GSE42743 oral cancer patients (**D**). The statistical significances were evaluated by paired *t*-test.

**Figure 2 cancers-12-02690-f002:**
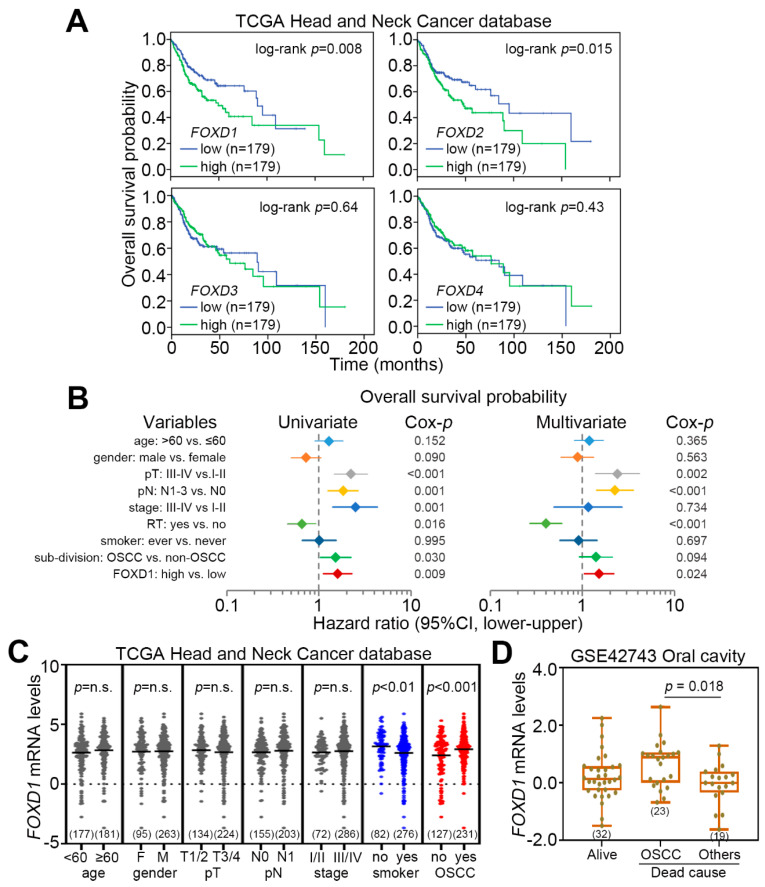
*FOXD1* serves as a poor prognostic marker in OSCC patients. (**A**) Kaplan–Meier analysis for *FOXD1*, *FOXD2*, *FOXD3* and *FOXD4* transcripts in TCGA head and neck cancer patients under the condition of overall survival probability. (**B**) Cox regression test using univariate and multivariate modes against the indicated variables, including age, gender, pathologic T stage (pT), pN, stage, radiation therapy (RT), smoking history, sub-division and *FOXD1* mRNA levels, under the condition of overall survival probability. CI denotes confidence interval. (**C**) Dot plot for *FOXD1* mRNA levels in primary tumors derived from TCGA head and neck cancer patients divided by various clinical parameters. The statistical differences were analyzed by student t-test. (**D**) Boxplot for the mRNA levels of *FOXD1* in primary tumors derived from GSE42743 oral cancer patients with different follow-up results. The statistical significance was analyzed by one-way ANOVA using Tukey’s test.

**Figure 3 cancers-12-02690-f003:**
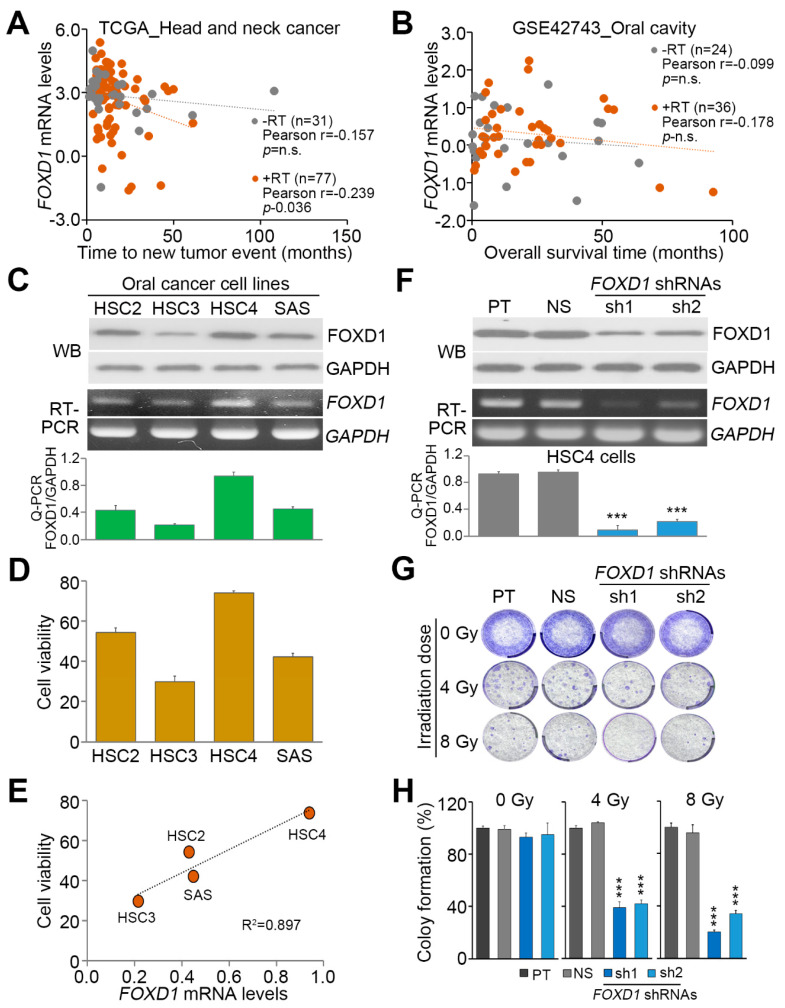
*FOXD1* upregulation predicts a poor response to radiotherapy in oral cancer. (**A**,**B**) Scatchard plot for *FOXD1* expression versus time to new tumor event in TCGA head and neck cancer patients (**A**) and *FOXD1* expression versus overall survival time in GSE42743 oral cancer patients (**B**) after radiation therapy. The Pearson correlation test was used to evaluate the statistical significance. (**C**) The protein levels detected by Western blot analyses (upper) and the mRNA levels detected by RT-PCR (middle) Q-PCR (lower) of FOXD1 and GAPDH in a panel of oral cancer cell lines HSC-2, HSC-3, HSC-4 and SAS. (**D**) Cell viability of the indicated oral cancer cell lines at 24 h post-exposure to 8 Gy irradiation. (**E**) Scatchard plot for the correlation between *FOXD1* expression and cell viability at 24 h post-exposure to 8 Gy irradiation in the detected oral cancer cell lines. (**F**) The protein levels detected by Western blot analyses (upper) and the mRNA levels detected by RT-PCR (middle) Q-PCR (lower) of FOXD1 and GAPDH in parental (PT) HSC4 cells and HSC4 cells transfected with non-silencing (NS) control shRNA or 2 independent *FOXD1* shRNAs. In **C** and **F**, GAPDH used as an internal control of the designated experiments. (**G**,**H**) Crystal violet staining for the cell colonies of HSC4 cell variants at 2 weeks post-exposure to the designated dose of irradiation (**G**) and the histograms for the results obtained from three independent experiments of colony-forming assay (**H**). In **C**, **F** and **H**, the error bars denote the data from three independent experiment presented as mean ± SEM. Non-parametric Friedman test was used to estimate the statistical significances. * *p* value < 0.05, ** *p* value < 0.01, *** *p* value < 0.001.

**Figure 4 cancers-12-02690-f004:**
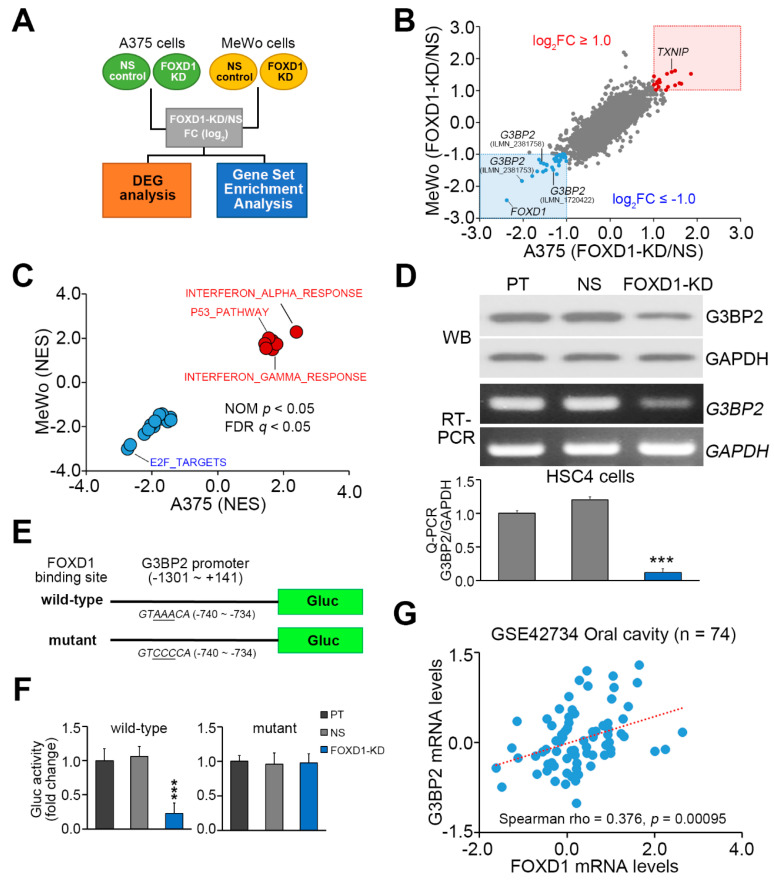
*FOXD1* knockdown induces the downregulation of *G3BP2* expression in oral cancer cells. (**A**) The flow-chart of processing and analyzing the microarray data from GSE111766 dataset. (**B**,**C**) Scatchard plot for the fold change (FC) of *FOXD1* mRNA levels in A375 and MeWo cells (**B**) and the normalized enrichment score (NES) of GSEA experiments. In **B**, the genes with log_2_FC ≥ 1.0 or ≤ −1.0 in A375 and MeWo cells after *FOXD1* knockdown are labeled as red and blue dots, respectively. In **C**, the significantly (nominal (NOM) *p* < 0.05 and false discovery rate (FDR) *p* < 0.05) upregulated (red dots) and downregulated (blue dots) gene sets in the A375 and MeWo cells after *FOXD1* knockdown. (**D**) The protein levels detected by Western blot analyses (upper) and the mRNA levels detected by RT-PCR (middle) and Q-PCR (lower) of *G3BP2* and GAPDH in parental (PT) HSC4 cells and HSC4 cells transfected with non-silencing (NS) control shRNA or *FOXD1* shRNA. GAPDH was used as an internal control of the designated experiments. (**E**,**F**) The constructs of Gaussia luciferase gene adjacent to the *G3BP2* promoter harboring wild-type or mutated Foxd1 DNA-binding sequences (**D**) and the histograms for the results of luciferase-based promoter activity assay in HSC4 cell variants. In **D** and **F**, the non-parametric Friedman test was used to estimate the statistical significances. (**G**) Scatchard plot for the expression of *FOXD1* and *G3BP2* in the primary tumors from GSE42734 oral cancer patients. Spearman correlation test was used to evaluate the statistical significance. * *p* value < 0.05, ** *p* value < 0.01, *** *p* value < 0.001.

**Figure 5 cancers-12-02690-f005:**
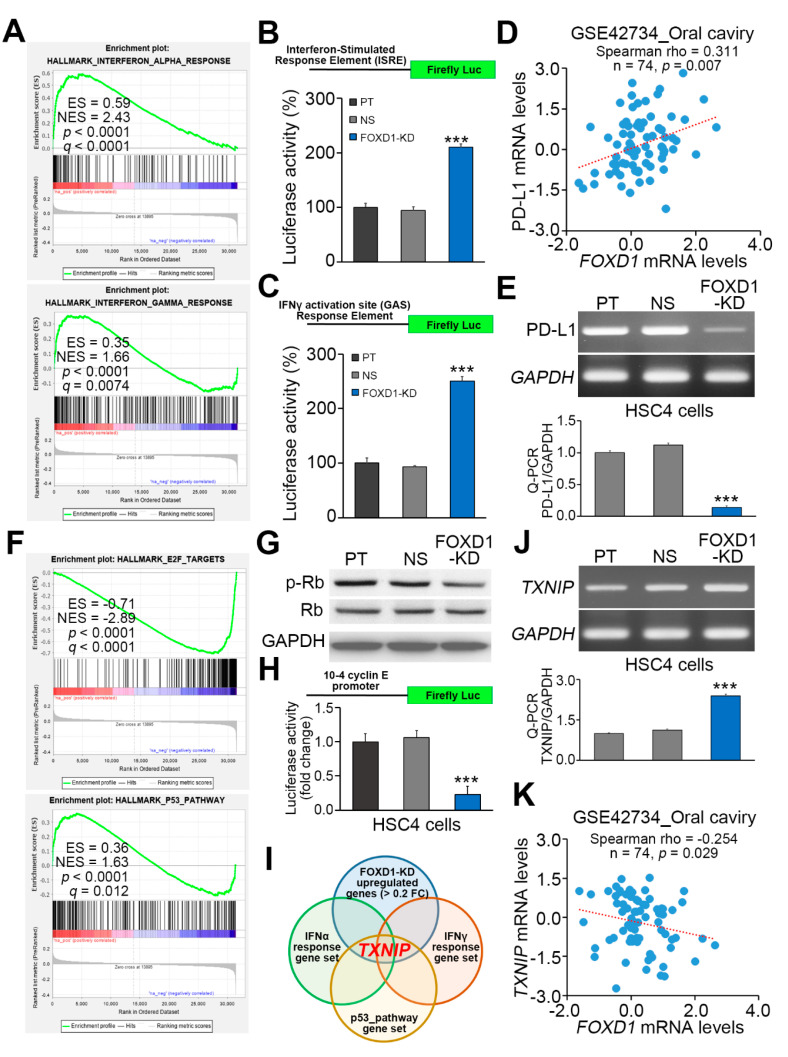
*FOXD1* knockdown induces the downregulation of the E2F signaling axis and the upregulation of *TXNIP*-associated interferon responses and p53 activation in oral cancer cells. (**A**) The enrichment score (ES) derived from the correlation among the IFN-α/γ-response gene sets and the queried Pearson’s correlation coefficient (r) is plotted (green curve). FDR denotes the false discovery rate. (**B**,**C**) The constructs of firefly luciferase gene adjacent to the IFN-stimulated response element (ISRE) and IFN-gamma activation site (GAS) response element (upper inserts) and the histograms for the results of luciferase-based promoter activity assays in HSC4 cell variants. (**D**) Scatchard plot for the expression of *FOXD1* and PD-L1 gene (*CD274*) in the primary tumors from GSE42734 oral cancer patients. (**E**) The mRNA levels of PD-L1 detected by RT-PCR (upper) and Q-PCR (lower) in parental (PT) HSC4 cells and HSC4 cells transfected with non-silencing (NS) control shRNA or *FOXD1* shRNA. (**F**) The enrichment score (ES) derived from the correlation among the E2F target/p53 pathway gene sets and the queried Pearson’s correlation coefficient (r) is plotted (green curve). (**G**) Western blot analyses for phosphorylated Rb, total Rb and GAPDH protein in HSC4 cell variants. GAPDH was used as an internal control of protein loading. (**H**) The constructs of firefly luciferase gene adjacent to the 10-4 cyclin E promoter (upper insert) and the histograms for the results of luciferase-based promoter activity assays in HSC4 cell variants. (**I**) The illustration for that *TXNIP* gene is included in the upregulated genes after FOXD1 knockdown, IFN-α/γ-response gene sets and p53 pathway gene set. (**J**) The mRNA levels of *TXNIP* detected by RT-PCR (upper) Q-PCR (lower) in parental (PT) HSC4 cells and HSC4 cells transfected with non-silencing (NS) control shRNA or *FOXD1* shRNA. (**K**) Scatchard plot for the expression of *FOXD1* and *TXNIP* in the primary tumors from GSE42734 oral cancer patients. In **D** and **K**, Spearman correlation test was used to evaluate the statistical significance. In **B**, **C**, **E**, **H** and **J**, non-parametric Friedman test was used to estimate the statistical significances. * *p* value < 0.05, ** *p* value < 0.01, *** *p* value < 0.001.

**Figure 6 cancers-12-02690-f006:**
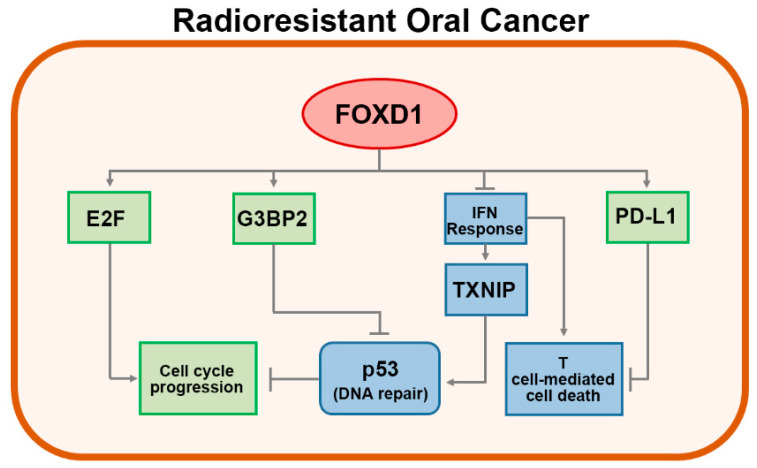
A possible mechanism for the *FOXD1*-promoted radioresistance in oral cancer cells. *FOXD1* forces cell cycle progression through the induction of E2F activation and *G3BP2* expression which subsequently inhibits the p53-mediated DNA repair and suppresses T cell-mediated cell death via elevating the expression of PD-L1 in radioresistant oral cancer cells. On the other hand, *FOXD1* alleviates the function of interferon (IFN)-responsive pathways on promoting the *TXNIP*-mediated p53 activation and enhancing T cell-mediated cell death in radioresistant oral cancer cells.

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
