# Peer review of "FOXD1 Repression Potentiates Radiation Effectiveness by Downregulating G3BP2 Expression and Promoting the Activation of TXNIP-Related Pathways in Oral Cancer"

_cancers, 2020, doi:10.3390/cancers12092690_

Round 1

Reviewer 1 Report

In the manuscript entitled: “FOXD1 repression potentiates radiation effectiveness by downregulating G3BP2 expression and promoting the activation of TXNIP-related pathways in oral cancer”, the authors identified a druggable target to enhance the effectiveness of radiotherapy on oral cancer patients,

The authors found that the upregulatio of FOXD1, a gene encoding forkhead box d1 (Foxd1), is extensively detected in primary tumors compared to normal tissues and associated with a poor outcome in oral cancer patients receiving irradiation treatment. Moreover, showed that the level of FOXD1 transcript is causally relevant to the effective dosage of irradiation in a panel of oral cancer cell lines. The FOXD1 knockdown (FOXD1-KD) dramatically suppressed the colony-forming ability of oral cancer cells after irradiation treatment.

The authors concluded that FOXD1 targeting might potentiate the anti-49 cancer effectiveness of radiotherapy and promote immune surveillance on oral cancer.

Major comments:

In general, the idea and innovation of this study, regards the analysis of downregulating G3BP2 expression during oral cancer is interesting, because the role of biomarkers released during oral diseases is validated but further studies on this topic could be an innovative issue in this field could be open a creative matter of debate in literature by adding new information. Moreover, there are few reports in the literature that studied this interesting topic with this kind of study design.

The study was well conducted by the authors; However, there are some concerns to revise that are described below.

The introduction section resumes the existing knowledge regarding the important factor linked with oral cancer and oral diseases.

However, as the importance of the topic, the reviewer strongly recommends, before a further re-evaluation of the manuscript, to update the literature through read, discuss and must cites in the references with great attention all of those recent interesting articles, that helps the authors to better introduce and discuss the role of some others mediators and nutraceutical agents, such as ADMA and Malondyaldehyde involved during oral diseases and periodontitis: 1) Isola G, Alibrandi A, Currò M, Matarese M, Ricca S, Matarese G, Ientile R, Kocher T. Evaluation of salivary and serum ADMA levels in patients with periodontal and cardiovascular disease as subclinical marker of cardiovascular risk. J Periodontol. 2020 91;8: 1076-1084. doi: 10.1002/JPER.19-0446. 2) Isola G, Polizzi A, Iorio-Siciliano V, Alibrandi A, Ramaglia L, Leonardi R. Effectiveness of a nutraceutical agent in the non-surgical periodontal therapy: a randomized, controlled clinical trial. Clin Oral Investig. 2020 Jun 17. doi: 10.1007/s00784-020-03397-z. 3) Isola G, Polizzi A, Santonocito S, Alibrandi A, Ferlito S. Expression of Salivary and Serum Malondialdehyde and Lipid Profile of Patients with Periodontitis and Coronary Heart Disease. Int J Mol Sci. 2019 Dec 1;20(23):6061. doi: 10.3390/ijms20236061.

The authors should be better specified, at the end of the introduction section, the rational of the study and the aim of the case report. In the material and methods section, should better clarify who performed data collection and the subject selection.

The discussion section appears well organized with the relevant paper that support the conclusions, even if the authors should better discuss the relationship between periodontitis and oral cancer. The conclusion should reinforce in light of the discussions.

In conclusion, I am sure that the authors are fine clinicians who achieve very nice results with their adopted protocol. However, this study, in my view does not in its current form satisfy a very high scientific requirement for publication in this journal and requests a revision before a futher re-evaluation of the manuscript.

Minor Comments:

Abstract:

  • Better formulate the abstract section by better describing the aim of the study

Introduction:

  • Please refer to major comments

Discussion

  • Please add a specific sentence that clarifies the results obtained in the first part of the discussion
  • Page 11 last paragraph: Please reorganize this paragraph that is not clear

Author Response

Comments and Suggestions for Authors In the manuscript entitled: “FOXD1 repression potentiates radiation effectiveness by downregulating G3BP2 expression and promoting the activation of TXNIP-related pathways in oral cancer”, the authors identified a druggable target to enhance the effectiveness of radiotherapy on oral cancer patients, The authors found that the upregulation of FOXD1, a gene encoding forkhead box d1 (Foxd1), is extensively detected in primary tumors compared to normal tissues and associated with a poor outcome in oral cancer patients receiving irradiation treatment. Moreover, showed that the level of FOXD1 transcript is causally relevant to the effective dosage of irradiation in a panel of oral cancer cell lines. The FOXD1 knockdown (FOXD1-KD) dramatically suppressed the colony-forming ability of oral cancer cells after irradiation treatment. The authors concluded that FOXD1 targeting might potentiate the anti-49 cancer effectiveness of radiotherapy and promote immune surveillance on oral cancer. Major comments: In general, the idea and innovation of this study, regards the analysis of downregulating G3BP2 expression during oral cancer is interesting, because the role of biomarkers released during oral diseases is validated but further studies on this topic could be an innovative issue in this field could be open a creative matter of debate in literature by adding new information. Moreover, there are few reports in the literature that studied this interesting topic with this kind of study design. The study was well conducted by the authors; However, there are some concerns to revise that are described below. Response: Thank you so much for giving us these positive comments and constructive suggestions. We have point-by-point answered your questions in the following paragraphs and revised our manuscript in accordance with your suggestions.

The introduction section resumes the existing knowledge regarding the important factor linked with oral cancer and oral diseases. However, as the importance of the topic, the reviewer strongly recommends, before a further re-evaluation of the manuscript, to update the literature through read, discuss and must cites in the references with great attention all of those recent interesting articles, that helps the authors to better introduce and discuss the role of some others mediators and nutraceutical agents, such as ADMA and Malondyaldehyde involved during oral diseases and periodontitis: 1) Isola G,
Alibrandi A, Currò M, Matarese M, Ricca S, Matarese G, Ientile R, Kocher T. Evaluation of salivary and serum ADMA levels in patients with periodontal and cardiovascular disease as subclinical marker of cardiovascular risk. J Periodontol. 2020 91;8: 10761084. doi: 10.1002/JPER.19-0446. 2) Isola G, Polizzi A, Iorio-Siciliano V, Alibrandi A, Ramaglia L, Leonardi R. Effectiveness of a nutraceutical agent in the non-surgical periodontal therapy: a randomized, controlled clinical trial. Clin Oral Investig. 2020 Jun 17. doi: 10.1007/s00784-020-03397-z. 3) Isola G, Polizzi A, Santonocito S, Alibrandi A, Ferlito S. Expression of Salivary and Serum Malondialdehyde and Lipid Profile of Patients with Periodontitis and Coronary Heart Disease. Int J Mol Sci. 2019 Dec 1;20(23):6061. doi: 10.3390/ijms20236061. Response: According to these critical references, we have revised the Introduction and Discussion sections as follows and cited these important references as Ref. 52-54 in the revised manuscript.

Introduction (line 84-89 of the revision): …………….. Notably, the immune-inflammatory response has been considered to play a critical role in the malignant transformation of oral epithelium from patients with chronic periodontitis [30]. As a result, finding distinctions between oral cancer and periodontitis by saliva metabolites has been thought to be a quickly and effectively non-invasive method to aid clinicians in determining between oral cancer and periodontitis [30]. Nevertheless, the role of FOX family in regulating this malignant transformation remains unknown……………………………………

Discussion (line 339-353 of the revision): Chronic inflammation of periodontitis has been linked to an increased risk of oral cancer. Moreover, the interaction between inflammation and immune responses in periodontal tissues of patients with chronic periodontitis was found to promote the malignant transformation of oral epithelium. In addition, during periodontitis, it has been found that the regulatory CD4 T cells (Tregs) accumulate at the inflamed tissues to limit the immune responses. Therefore, the suppression of inflammation in patients with periodontitis by anti-inflammatory agent, such as nutraceutical drugs [52], might be usefulness for preventing tumorigenesis. Besides, targeting FOXD1 might be able to overcome the suppressed cytotoxic T cell functions at the oral cancer tissues surrounding with inflamed lesions since FOXD1 repression enhanced the IFNresponsive signaling pathways and reduced the expression of PD-L1 in oral cancer cells. On the other hand, the identification of salivary biomarkers to distinguish periodontitis from oral cancer was thought to be an important clinical issues. Recently, several salivary biomarkers, e.g. asymmetric dimethylarginine [53] and malondialdehyde [54], have been identified for periodontitis. Because FOXD1 expression in oral cancer tissues compared to normal tissues is predominantly upregulated, we thought that FOXD1 might be a potential biomarker to differentiate oral cancer from periodontitis.

The authors should be better specified, at the end of the introduction section, the rational of the study and the aim of the case report. In the material and methods section, should better clarify who performed data collection and the subject selection.
Response: As suggested, we have revised the last paragraph of the introduction section as shown in the follows to specify the rationale and the aim of the case report at the end of the introduction section (please see line 92-104 of the revision).

Original: This aim of this study was thus focused on investigating the clinical relevance of FOXD1 expression in oral cancer patients receiving radiotherapy and uncovering a possible mechanism in which FOXD1 promotes radioresistance in oral cancer. In this study, we firstly document that FOXD1 upregulation correlates with a poor responsiveness of oral cancer patients to radiotherapy and desensitizes oral cancer cells to irradiation treatment probably via elevating the G3BP2 and E2F-related pathways and suppressing the signaling cascades related to the TXNIP-associated interferon responsiveness and p53 activity.

Revised: Although previous reports have shown that FOXD subtypes (FOXDs) play a critical role in the mechanism for tumorigenesis and cancer progression, their roles in regulating oral cancer development and conferring the radiation resistance of oral cancer remain largely unknown. The aim of this study was thus focused on dissecting the transcriptional profiling of FOXDs in normal tissues and primary tumors and evaluating their clinical relevance in oral cancer. Our data demonstrated that the upregulation of FOXD1 compared to other FOXDs is extensively detected in primary tumors and significantly correlated with a poorer clinical outcome in oral cancer patients. Our results further showed that that FOXD1 upregulation correlates with a poor responsiveness of oral cancer patients to radiotherapy and desensitizes oral cancer cells to irradiation treatment probably via elevating the G3BP2 and E2F-related pathways and suppressing the signaling cascades related to the TXNIP-associated interferon responsiveness and p53 activity. This study is the first to document the oncogenic role of FOXD1 in oral cancer.

As suggested, we also revised the MM section as shown in the follows to clarify who performed data collection and the subject selection.

Original: The clinical data and overall survival (OS) time for TCGA head and neck cancer patients were collected from the UCSC Xena website (UCSC Xena. Available online: http://xena.ucsc.edu/welcome-to-ucsc-xena/). The molecular data obtained by RNAseq (polyA þ Illumina HiSeq) analysis of the TCGA head and neck cancer cohort were also downloaded from the UCSC Xena website. Microarray results with accession numbers GSE42734 and GSE111766 and the related clinical data were obtained from the Gene Expression Omnibus (GEO) database on the NCBI website. …………………….

Revised: The clinical data and overall survival (OS) time for TCGA head and neck cancer patients were collected from the UCSC Xena website (UCSC Xena. Available online: http://xena.ucsc.edu/welcome-to-ucsc-xena/). The molecular data obtained by RNAseq (polyA þ Illumina HiSeq) analysis of the TCGA head and neck cancer cohort were also downloaded from the UCSC Xena website. Microarray results with accession numbers GSE42743 which was performed by Holsinger C et. al. from Stanford University School of Medicine to compare differences of gene expression between oral cancer samples and adjacent normal mucosa and GSE111766 which was
established by Larribère L et. al. from DKFZ Research Center in Germany to dissect the alteration of gene expression after FOXD1 knockdown in melanoma cell lines, and the related clinical data were obtained from the Gene Expression Omnibus (GEO) database on the NCBI website. …………………….

The discussion section appears well organized with the relevant paper that support the conclusions, even if the authors should better discuss the relationship between periodontitis and oral cancer. The conclusion should reinforce in light of the discussions. Response: As suggested, we have discussed the relationship between periodontitis and oral cancer in the last paragraph of Discussion section of revision (please see line 339-353 of the revision). Please also see the content above. Besides, we have revised the conclusion as follows.

Original: Our results suggest that FOXD1 upregulation may involve in the mechanism for radioresistance in oral cancer. The targeting of FOXD1 might be a good strategy to enhance radiosensitivity of oral cancer via downregulating G3BP2-related pathways and upregulating the TXNIP-associated cellular functions.

Revised: Our results demonstrate that FOXD1 is highly expressed by primary tumors compared to the adjacent normal tissues and serves as a poor prognostic marker in oral cancer patients receiving irradiation therapy. The targeting of FOXD1 might be a good strategy to enhance radiosensitivity of oral cancer cells via downregulating G3BP2-related pathways and upregulating the TXNIP-associated cellular functions. However, further experiments are needed to delineate the role of FOXD1 in the malignant transformation of oral epithelium from patients with chronic periodontitis.

In conclusion, I am sure that the authors are fine clinicians who achieve very nice results with their adopted protocol. However, this study, in my view does not in its current form satisfy a very high scientific requirement for publication in this journal and requests a revision before a futher re-evaluation of the manuscript. Response: Thank you again for your positive comments to our manuscript.

Minor Comments: Abstract: Better formulate the abstract section by better describing the aim of the study Response: As suggested, we have clearly formulated the aim of the study in the Abstract section
as follow (please also see line 36-38 of the revision).

Original: Radiotherapy is commonly used to treat oral cancer patients in the current clinics; however, a subpopulation of patients shows a poor radiosensitivity. Therefore, identifying a druggable target to enhance the effectiveness of radiotherapy on oral cancer patients is urgently needed. By performing an in silico analysis ………………………….

Revised: Radiotherapy is commonly used to treat oral cancer patients in the current clinics; however, a subpopulation of patients shows a poor radiosensitivity. Therefore, the aim of this study was to identify a biomarker or druggable target to enhance the therapeutic effectiveness of radiotherapy on oral cancer patients. By performing an in silico analysis …………………………. Introduction: Please refer to major comments Response: As suggested in major comments, we have revised the Introduction (please see the response to the related major comment above and line 84-89 of the revision)

Discussion Please add a specific sentence that clarifies the results obtained in the first part of the discussion Response: As requested, we have added a sentence “Therefore, the therapeutic targeting of FOXD1 might be a new strategy to potentiate the efficacy of irradiation in treating oral cancer” at the end of the first part of the discussion (please also see line 303-304 of the revision) to clarify the results obtained in this study.

Page 11 last paragraph: Please reorganize this paragraph that is not clear Response: As suggested, we have revised this paragraph as follows (please also see line 305-316 of the revision).

Original: Radiation therapy was previously thought to correlate with the induction of an intratumoral type I interferon (IFN) response [44]. As a result, several clinical trials were performed to evaluate the combination of IFN therapy with radiation therapy, owing to the radiosensitizing effect of interferons [45-47]. Similarly, in this study, our results demonstrated that FOXD1 knockdown enhances the anti-cancer effectiveness of irradiation via activating the pathway of IFN-, as well as IFN-, response in oral cancer cells. On the other hand, the immunostimulatory effects of radiation therapy, such as improved immune cell recruitment and enhanced susceptibility to T cellmediated cell death, were reported previously [48]. Here we further find that FOXD1 knockdown is concurrently accompanied with a reduced expression of PD-L1 which is a critical suppressor for T cell function through the binding with PD-1 in oral cancer
cells with a poorer radiosensitivity. These findings suggest that FOXD1 repression may not only promote radiosensitivity but also potentiate the T cell-mediated adaptive immunity in oral cancer.

Revised: Radiation therapy has been shown to enhance the anti-tumor capacity of adaptive immunity by augmenting a type I interferon (IFN)-dependent innate immune sensing of tumors [45]. Based on this immunomodulatory effect of radiation therapy, several clinical trials were performed to evaluate the anti-cancer effectiveness of combining irradiation with IFNs [46-48]. Similarly, our results demonstrated that FOXD1 knockdown enhances the radiosensitivity of oral cancer cells via activating the of IFN and IFN--responsive pathways. In addition, the immunostimulatory effects of radiation therapy, such as improved immune cell recruitment and enhanced susceptibility to T cell-mediated cell death, were also reported previously [49]. Here we found that FOXD1 knockdown is concurrently accompanied with a reduced expression of PD-L1, a critical suppressor for T cell function through the binding with PD-1, in oral cancer cells with a poorer radiosensitivity. These findings suggest that FOXD1 repression may not only promote therapeutic efficacy of irradiation but also potentiate the T cell-mediated adaptive immunity in combating oral cancer.

Reviewer 2 Report

In this manuscript, by  analyzing  public TCGA data base for  Head and Neck Cancer and data base for oral cancer Lin and colleagues have shown  that gene encoding forkhead box d1 (Foxd1), is significantly detected in primary tumors compared to adjacent normal tissues. They have also shown that knockdown  of FOXD1 in oral cancer cell line (HSC4) dramatically suppresses the colony-forming ability of oral cancer cells after irradiation treatment. Further, by DEG analysis they have shown that G3BP2, a negative regulator of p53, is predominantly repressed after FOXD1-KD and transcriptionally regulated by Foxd1. Overall the study has significance. However, the study has several limitations- its mostly based on already published database. The rationale of some of the experiments/cell line used/data base used are not properly delineated. Some of the analysis lacks statistical power- and hence need additional support. The study also need some additional experiments/more stringent analysis of the data.

MAJOR:

  1. FOXD2 and FOXD4 is significantly upregulated in primary tumors compared to normal adjacent tissues derived from HNC subjects but not from oral cancer patients- Any explanations?
  2. For correlation (cell viability vs FOXD1 mRNA level) only four cell lines were used (Fig 3E)- In terms of statistical power -this seems to be very low numbers for "correlation analysis".
  3. It would also be wise to include the viability data for sh1 and sh2 cells (Fig 3F) in the correlation analysis for Fig 3E. Similarly, it would be wise to include CFU data (as shown in Fig 3G/H) for other cell lines: specifically, HSC3, the levels of FOXD1 mRNA is close to sh1 and sh2 derived from HSC4. So, it would be prudent to generate comparative data for HSC3 and sh1/sh2.
  4. The Western data for FOXD1 knockdown cell lines also needs to be included. For figure 3C,3F and 4D.
  5. Section 2.4: Rationale of using data from A375 and MeWo melanoma cells in the context of oral cancer is not clear! How about microarray data comparison for wild type HSC4 with FOXD1 KD cells? This would be more relevant for the study and it would strengthen the overall premises of the study.
  6. Fig 4B: What is the rationale of comparing (Fold change values) between two cell lines?
  7. “G3BP2 downregulation is predominant in HSC4 cells after FOXD1 knockdown” is it true for other OSCC cell lines? Expression of FOXD1 is less in HSC3 cells compared to HSC4-how about expression of G3BP2 in HSC3? How the expression of G3BP2 correlates with FoxD1 in different cell lines (e.g. HSC2, 3, 4 and Sas used in the study)?
  8. What was the p value cutoff for FC values in figure 4B and Supplementary Table 1? In sup table 1, the FC values are very low (are those log2 values??) ….Not sure whether those are significant!! e.g. FC value for G3BP2 (the main target of the study ) is 0.28 –are not reliable!! (main drawbacks). FDR adjusted p-value needs to be included for all the genes listed in supplementary table 1.
  9. “FOXD1 and G3BP2 is positively correlated in primary tumors derived from GSE42734 OSCC patients” – the spearman rho is 0.37 – Is not a good correlation. How about this correlation in TCGA database (for cancerous tissue and normal tissues, independently)?

MINLOR

  1. Section 2.2: FOXD1 upregulation predicts a poor in oral cancer patients: what does it mean by predicts a poor??
  2. patients dead from OSCC: not right
  3. Line 209: “To under if the..” – please check the sentence
  4. Fig 3 legend: Cell viability of the detected oral cancer cells: what does it mean by detected oral cancer cells? Detected cells have been used in several places!!
  5. Why G3BP2 listed three times in the insert of Fig 4B?
  6. What were the sources of the cell lines (HSC2,3,4 and Sas) used? Please cite references for those cell lines. Are those ATCC cell lines? are those authenticated?

Author Response

In this manuscript, by analyzing public TCGA data base for Head and Neck Cancer and data base for oral cancer Lin and colleagues have shown that gene encoding forkhead box d1 (Foxd1), is significantly detected in primary tumors compared to adjacent normal tissues. They have also shown that knockdown of FOXD1 in oral cancer cell line (HSC4) dramatically suppresses the colony-forming ability of oral cancer cells after irradiation treatment. Further, by DEG analysis they have shown that G3BP2, a negative regulator of p53, is predominantly repressed after FOXD1-KD and transcriptionally regulated by Foxd1. Overall the study has significance. However, the study has several limitations- its mostly based on already published database. The rationale of some of the experiments/cell line used/data base used are not properly delineated. Some of the analysis lacks statistical power- and hence need additional support. The study also need some additional experiments/more stringent analysis of the data. Response: Thank you very much for giving us the positive comments and pointing out our weaknesses. According to your constructive suggestions, we have performed new experiments and re-analyzed the in silico experiments in the revised manuscript, and point-by-point answered your questions in the following paragraphs.

MAJOR: 1. FOXD2 and FOXD4 is significantly upregulated in primary tumors compared to normal adjacent tissues derived from HNC subjects but not from oral cancer patients- Any explanations? Response: Thank you for this critical comment. Indeed, FOXD2 and FOXD4, similar to FOXD1, is significantly upregulated in primary tumors compared to normal adjacent tissues derived from TCGA head and neck cancer subjects but not from GSE42743 oral cancer patients. To answer this question, we dissect the transcriptional profile of FOXD1, FOXD2 and FOXD4 in the anatomic subdivision of TCGA head and neck cancer. As shown in the following figures (A and B), compared to normal tissues and oral cavity, the mRNA levels of FOXD2 in hypopharynx and FOXD4 in tonsil are significantly upregulated. Whereas FOXD4 expression shows no difference in the paired normal adjacent tissue (NAT) and primary tumor derived from TCGA oral cancer patients, the mRNA levels of FOXD1 and FOXD2 in tumors are significantly higher than that of NAT (C). Nevertheless, the mean of log2 fold change for FOXD1 expression (2.556) is relatively higher than that for FOXD2 expression (1.374) in tumor compared to NAT (C). Therefore, the significant upregulation of FOXD2 and
FOXD4 in primary tumor derived from TCGA HNC subjects might be closely associated with the development of hypopharynx and tonsil cancer, respectively. We have added these results as Supplementary Figure 1 and provided the descriptions in Results section of revised manuscript (please see line 119-127 of the revision).

Supplementary Figure 1. The transcriptional profile of FOXD1, FOXD2 and FOXD4 in the anatomic subdivision of TCGA head and neck cancer. (A and B) The heatmap (A) and boxplot (B) for the mRNA levels of FOXD1, FOXD2 and FOXD4 in the normal tissues and primary tumors derived from the different anatomic subdivision of TCGA head and neck cancer. The statistical differences were analyzed by student t-test. (C) The mRNA levels of FOXD1, FOXD2 and FOXD4 in the normal adjacent tissues (NAT) and primary tumors derived from oral cavity using TCGA head and neck cancer database. The statistical significances were evaluated by paired t-test.

2. For correlation (cell viability vs FOXD1 mRNA level) only four cell lines were used (Fig 3E)- In terms of statistical power -this seems to be very low numbers for "correlation analysis". Response: Thank you for this comment. Exactly, we agree with your point that only four cell lines seems to be very low numbers to perform correlation analysis. As a result, in
the Figure 3E, we used the R2 value of liner regression to implicate a potentially positive correlation between FOXD1 expression and cell viability after RT exposure in the 4 oral cancer cell lines. We replaced “positively correlated” with “causally associated” in the revised manuscript (please see line 168 of the revision).

3. It would also be wise to include the viability data for sh1 and sh2 cells (Fig 3F) in the correlation analysis for Fig 3E. Similarly, it would be wise to include CFU data (as shown in Fig 3G/H) for other cell lines: specifically, HSC3, the levels of FOXD1 mRNA is close to sh1 and sh2 derived from HSC4. So, it would be prudent to generate comparative data for HSC3 and sh1/sh2. Response: As requested, we have included the viability data for sh1 and sh2 cells as Supplementary Figure 3 (please see the following Figure) and added a description of data in Results section of the revised manuscript (please see line 174 of the revision).

Supplementary Figure 3. FOXD1 knockdown sensitizes HSC4 cells to irradiation treatment. Cell viability of parental (PT) HSC4 cells and HSC4 cells transcfected with non-silencing (NS) control shRNA or 2 independent FOXD1 shRNAs at 24 hours postexposure to 8 Gy irradiation. The data obtained from three independent experiment presented as mean ± SEM. Non-parametric Friedman test was used to estimate the statistical significances. The symbol “***” denotes statistical p value < 0.001.

In addition, we have also included CFU data for HSC2, HSC3, HSC4 and SAS oral cancer cells as Supplementary Figure 2 (please see the Figure below) and added a description of data in Results section of the revised manuscript (please see line 169 of the revsion).

Supplementary Figure 2. The colony-forming ability of HSC2, HSC3, HSC4 and SAS oral cancer cells after irradiation exposure at the designated doses. (A and B) Crystal violet
staining for the cell colonies of HSC2, HSC3, HSC4 and SAS oral cancer cells at 2 weeks post-exposure to the designated dose of irradiation (A) and the histograms for the results obtained from three independent experiments of colony-forming assay (B).

4. The Western data for FOXD1 knockdown cell lines also needs to be included. For figure 3C,3F and 4D. Response: As requested, we have provided the Western blot results in the revised Figure 3C, 3F and 4D (please see the following figures) and descriptions in Results section (please see line168, 173 and 205 of the revision).

New Figure 3C and 3F:

New Figure 4D:

5. Section 2.4: Rationale of using data from A375 and MeWo melanoma cells in the context of oral cancer is not clear! How about microarray data comparison for wild type HSC4 with FOXD1 KD cells? This would be more relevant for the study and it would strengthen the overall premises of the study. Response: Thank you for this critical comment. In this study, we used GSE111766 dataset which contains microarray results for the control and FOXD1-knockdown A375 and MeWo melanoma cells as a discovery set to identify the potentially FOXD1-regualted genes
and pathways by DEG analysis and GSEA simulation, respectively. We then validated the candidates obtained from these 2 in silico analyses in the HSC4 oral cancer cells without and with FOXD1 knockdown. We agree with your point that using the microarray data comparison for wild type HSC4 with FOXD1 KD cells would be more relevant for the study and strengthen the overall premises of the study. Because we did not find any microarray experiments regarding FOXD1 KD oral cancer cells in the Gene Expression Omnibus (GEO) database, we thus alternatively decided to re-analyze the microarray data of GSE111766 dataset. Moreover, radiation therapy is also commonly used to destroy the remaining melanoma after surgery even though melanoma and oral cancer are different cancer types. Besides, overcoming the radioresistance is also a critical issue in treating melanoma patients [Cutaneous Melanoma: Etiology and Therapy (2017); https://www.ncbi.nlm.nih.gov/books/NBK481863/#chapter8.s1].

6. Fig 4B: What is the rationale of comparing (Fold change values) between two cell lines? Response: Thank you for this good question. The reasons of performing DEG analysis in A375 and MeWo cell lines are as follows. 1. To ensure the efficiency of FOXD1 knockdown in two cell lines. In Figure 4B, we found that the fold-change value of FOXD1 expression is the top 1 in the downregulated genes of two cell lines after FOXD1 knockdown. 2. To verify the candidates from DEG analysis are indeed concurrently upregulated or downregulated in two cells, thereby reducing the off-target effects.

7.“G3BP2 downregulation is predominant in HSC4 cells after FOXD1 knockdown” is it true for other OSCC cell lines? Expression of FOXD1 is less in HSC3 cells compared to HSC4-how about expression of G3BP2 in HSC3? How the expression of G3BP2 correlates with FoxD1 in different cell lines (e.g. HSC2, 3, 4 and Sas used in the study)? Response: Thank you for these critical comments. The description regarding “G3BP2 downregulation is predominant in HSC4 cells after FOXD1 knockdown” seems to be not proper. We have revised this sentence to be “FOXD1 knockdown dramatically reduces the protein and mRNA levels in HSC4 cells” in the revised manuscript (please see line 205 of the revision). In addition, as requested, we performed another RT-PCR and Q-PCR experiment to measure the mRNA levels of G3BP2 in 4 oral cancer cell lines (please see the following figure). The data showed that the mRNA levels of G3BP2, similar to FOXD1, in HSC4 cells are relatively higher than other oral cancer
cell lines. We have added the result as Supplementary Figure 4 and provided the description in Results section (please see line 206-207 of the revision) of the revised manuscript.

Supplementary Figure 4. The measurement of G3BP2 mRNA levels in oral cancer cell lines. The mRNA levels of G3BP2 and GAPDH detected by RT-PCR and Q-PCR experiments in a panel of oral cancer cell lines HSC-2, HSC-3, HSC-4 and SAS. GAPDH was used as an internal control of RT-PCR and Q-PCR experiment.

8. What was the p value cutoff for FC values in figure 4B and Supplementary Table 1? In sup table 1, the FC values are very low (are those log2 values??) ….Not sure whether those are significant!! e.g. FC value for G3BP2 (the main target of the study ) is 0.28 –are not reliable!! (main drawbacks). FDR adjusted p-value needs to be included for all the genes listed in supplementary table 1. Response: Thank you for your critical comments. In the original Figure 4B and Table S1, we used the log2FC values of GSE111766 dataset derived from the quantile normalization of IlluminaGUI in R. The data processing seems to be not proper. In this revised manuscript, we utilized GEO2R program to obtain the log2FC values and FDR adjusted p-value and revise the original Figure 4B and Table S1 (please the new Figure 4B and Table S1 below). By using the log2FC values from GEO2R, we also revised the Figure 4C and Table S2 (please the new Figure 4C and Table S2 below) by performing a new GSEA experiment. We have also provided the method for data processing in MM section of the revised manuscript (please see the descriptions below).

MM section: 4.1. Data collection and processing from TCGA and GEO databases The clinical data and overall survival (OS) time for TCGA head and neck cancer patients were collected from the UCSC Xena website (UCSC Xena. Available online: http://xena.ucsc.edu/welcome-to-ucsc-xena/). …………………………………… The fold changes of gene expression after FOXD1 knockdown in A375 [FOXD1 knockdown #1 (GSM3039523, GSM3039524, GSM3039525) versus control (GSM3039516, GSM3039517, GSM3039518) and MeWo (FOXD1 knockdown #1 (GSM3039519, GSM3039520) versus control (GSM3039514, GSM3039515) were obtained by using GEO2R software and presented as log2 values.
New Figure 4B:

New Table S1:

New Figure 4C:

New Table S2:

9.“FOXD1 and G3BP2 is positively correlated in primary tumors derived from GSE42734 OSCC patients” – the spearman rho is 0.37 – Is not a good correlation. How about this correlation in TCGA database (for cancerous tissue and normal tissues, independently)? Response: As suggested, we dissected the transcriptional profile of FOXD1 and G3BP2 in cancerous and normal tissues derived from TCGA head and neck cancer (HNC) database. As shown in the following figures, the correlation between FOXD1 and G3BP2 mRNA levels appears to be negatively correlated in normal tissues, HNC
primary tumors and cancerous tissues from oral cavity even though a statistical significance (p = 0.00031) was only shown in HNC primary tumors. Because cancerous tissues from oral cavity used in GSE42743 (GSE42734 in not correct. we have revised this typo error in the revised manuscript) are HPV-negative samples, we further analyzed the correlation between FOXD1 and G3BP2 mRNA levels in the cancerous tissues from TCGA oral cavity samples detected to be HPV-negative. Although the statistical power is not strong due to a limited sample size, we did find a positive correlation between FOXD1 and G3BP2 mRNA levels. We provided the Sample ID of TCGA HNC database for these HPV-negative tumors of oral cavity to easily check this result. Therefore, based on these findings, the results of HPV test might affect the co-expression between FOXD1 and G3BP2 in oral cancer. The explanation of a poor correlation in GSE42734 OSCC patients is that an oncogenic stimulation inducing the transcription factor activity of FOXD1 might be also needed to force the expression of G3BP2 during the malignant evolution of oral cancer.

MINLOR 10. Section 2.2: FOXD1 upregulation predicts a poor in oral cancer patients: what does it mean by predicts a poor?? Response: Thank you pointing out this mistake. It should be “FOXD1 upregulation predicts a poor prognosis in oral cancer patients”. We have revised this mistake in the revised manuscript (please see line 137 of the revision).
11. patients dead from OSCC: not right Response: Thank you for pointing out this mistake. It should be “patients death from OSCC”. Please see line 149 of the revision.

12. Line 209: “To under if the..” – please check the sentence Response: Thank you for pointing out this mistake. We have revised the sentence to be “To validate if the ………”. Please see line 241 of the revision.

13. Fig 3 legend: Cell viability of the detected oral cancer cells: what does it mean by detected oral cancer cells? Detected cells have been used in several places!! Response: As suggested, we have revised the sentence to be “Cell viability of the indicated oral cancer cell lines”. We also replaced “detected cells” with “A375 and MeWo cells” in Figure 4 legend.

14. Why G3BP2 listed three times in the insert of Fig 4B? Response: The log2FC values of three G3BP2 gene symbols were detected by its three independent probes in Illumina HumanHT-12 V4.0 expression beadchip. In the revised Figure 4B, we have provided the different probe IDs of G3BP2 gene (Please see the following figure).

15. What were the sources of the cell lines (HSC2,3,4 and Sas) used? Please cite references for those cell lines. Are those ATCC cell lines? are those authenticated?
Response: We are sorry for this mistake. The sources of the cell lines (HSC2,3,4 and Sas) used in this study were obtained from Japanese Collection of Research Bioresources (JCRB) Cell Bank, not ATCC. We have revised this mistake in the revised manuscript (please see line xxx). The order information for co-author Wei-Min Chang is shown as below. We have used these cell lines to perform the in vitro and in vivo experiments in our previous report (Chien MH et. al., Mol Cancer Ther. 2017 Jun;16(6):1102-1113).

Round 2

Reviewer 1 Report

In the R1 version of the manuscript entitled: “FOXD1 repression potentiates radiation effectiveness by downregulating G3BP2 expression and promoting the activation of TXNIP-related pathways in oral cancer” the authors followed all the issues suggested by the reviewer. Though the changes based on the reviewer comments, almost of the criticisms were carefully analysed and solved.

I have carefully evaluated all parts of the manuscript. I believe that the article, in this version, is now adequate for publication in this journal.

Author Response

In the R1 version of the manuscript entitled: “FOXD1 repression potentiates radiation effectiveness by downregulating G3BP2 expression and promoting the activation of TXNIP-related pathways in oral cancer” the authors followed all the issues suggested by the reviewer. Though the changes based on the reviewer comments, almost of the criticisms were carefully analysed and solved.

I have carefully evaluated all parts of the manuscript. I believe that the article, in this version, is now adequate for publication in this journal.

Response:

Thank you so much for your positive comments and critical suggestions.

Reviewer 2 Report

The authors satisfactorily improved the manuscript.

Author Response

The authors satisfactorily improved the manuscript.

Response:

Thank you so much for your critical comments and suggestions.